# PI3K Inhibitors for the Treatment of Chronic Lymphocytic Leukemia: Current Status and Future Perspectives

**DOI:** 10.3390/cancers14061571

**Published:** 2022-03-18

**Authors:** Iwona Hus, Bartosz Puła, Tadeusz Robak

**Affiliations:** 1Department of Hematology, Institute of Hematology and Transfusion Medicine, 02-776 Warsaw, Poland; ihus@ihit.waw.pl (I.H.); bpula@ihit.waw.pl (B.P.); 2Copernicus Memorial Hospital, 93-510 Lodz, Poland; 3Department of Hematology, Medical University of Lodz, 93-510 Lodz, Poland

**Keywords:** BGB-10188, chronic lymphocytic leukemia, copanlisib, duvelisib, idelalisib, parsaclisib, PI3-kinase inhibitors, treatment, safety, umbralisib, zandelisib

## Abstract

**Simple Summary:**

The development of small agents targeting the B-cell receptor (BCR) pathway revolutionized the treatment of chronic lymphocytic leukemia (CLL). BCR-dependent leukemic cell proliferation is governed by phosphoinositide 3-kinase (PI3K) signaling. The selective PI3Kδ inhibitor idelalisib and dual PI3Kδ/γ inhibitor duvelisib are currently approved by the Food and Drug Administration and European Medicine Agency (only idelalisib) for CLL treatment. Umbralisib, a selective PI3Kδ and casein kinase-1ε (CK1ε) inhibitor, has a different chemical structure and a more favorable safety profile than other PI3K inhibitors (PiK3is); this has enabled its use in combination regimens in clinical trials in first-line and relapsed/refractory CLL. This paper summarizes the development of PI3Kis, their current role and future perspectives in the treatment of patients with CLL.

**Abstract:**

Phosphoinositide 3-kinases (PI3Ks) signaling regulates key cellular processes, such as growth, survival and apoptosis. Among the three classes of PI3K, class I is the most important for the development, differentiation and activation of B and T cells. Four isoforms are distinguished within class I (PI3Kα, PI3Kβ, PI3Kδ and PI3Kγ). PI3Kδ expression is limited mainly to the B cells and their precursors, and blocking PI3K has been found to promote apoptosis of chronic lymphocytic leukemia (CLL) cells. Idelalisib, a selective PI3Kδ inhibitor, was the first-in-class PI3Ki introduced into CLL treatment. It showed efficacy in patients with del(17p)/*TP53* mutation, unmutated IGHV status and refractory/relapsed disease. However, its side effects, such as autoimmune-mediated pneumonitis and colitis, infections and skin changes, limited its widespread use. The dual PI3Kδ/γ inhibitor duvelisib is approved for use in CLL patients but with similar toxicities to idelalisib. Umbralisib, a highly selective inhibitor of PI3Kδ and casein kinase-1ε (CK1ε), was found to be efficient and safe in monotherapy and in combination regimens in phase 3 trials in patients with CLL. Novel PI3Kis are under evaluation in early phase clinical trials. In this paper we present the mechanism of action, efficacy and toxicities of PI3Ki approved in the treatment of CLL and developed in clinical trials.

## 1. Introduction

Chronic lymphocytic leukemia (CLL) is a lymphoproliferative neoplasm deriving from B lymphocytes. While leukemic cells morphologically resemble mature B cells, their immune function is impaired. CLL cells are also characterized by a prolonged life span due to overexpression of antiapoptotic proteins, mainly B cell lymphoma 2 (Bcl-2) and inhibitor of apoptosis (IAP) families [1]. Malignant transformation might occur in the B cells before or after somatic hypermutation of the genes encoding the immunoglobulin heavy chain variable (IGHV) region. IGHV mutational status has both significant prognostic [2] and predictive value, with unmutated IGHV being associated with shorter response to treatment with immunochemotherapy [3,4]. The most unfavorable outcome in the terms of overall survival (OS) is associated with 17p deletion and/or *TP53* mutation, mainly due to refractoriness to chemo- and chemoimmunotherapy with anti-CD20 monoclonal antibodies [5].

Immunochemotherapy has been the mainstay of CLL treatment since 2008, when the results of a CLL8 trial showed improved response rates and better progression-free survival (PFS) and OS when the anti-CD20 monoclonal antibody rituximab was added to fludarabine and cyclophosphamide (FC) chemotherapy [6]. However, this benefit did not include patients with 17p deletion/*TP53* mutation or with unmutated IGHV [4,6]. As such, both patient groups require treatment other than immunochemotherapy, starting from the first line of therapy.

For a long time, poor outcome was also associated with a lack, or short, response to immunochemotherapy. A better understanding of the biology of CLL therapy, particularly regarding the role of B cell receptor (BCR) signaling for leukemic cell survival and proliferation, has resulted in better treatment options [7]. Following this, small molecule agents known to inhibit BCR and antagonize the anti-apoptotic protein Bcl-2 have become the standard treatment options in CLL [8].

There are two groups of drugs that target BCR in CLL patients: Bruton’s tyrosine kinase (BTK) inhibitors (BTKis) and phosphoinositide 3-kinase inhibitors (PI3Kis) [9,10]. BTKis are widely used in any treatment line while PI3Kis are much less common, mainly due to the toxicities of the first-in-class inhibitor, idelalisib. However, despite significant progress in the design of therapy, CLL remains an incurable disease and new treatment options for relapsed/refractory (R/R) patients are still needed.

PI3K inhibitors constitute a promising group of small molecule agents and have been extensively studied in many cancers including B-cell malignancies, as both monotherapy and various combinations. A list of PI3Kis approved and evaluated in clinical trials in patients with CLL is presented in Table 1. In this review paper we present the mechanism of action, efficacy and toxicities of PI3Kis approved in the treatment of CLL and developed in clinical trials.

## 2. PI3K Pathway in Normal B Lymphocytes and CLL Cells

The PI3K kinases can be divided into three classes; class I is the most important in regard to lymphocyte biology and CLL pathogenesis. The class I PI3Ks are responsible for the phosphorylation of the phosphatidylinositol 4,5-bisphosphate (PIP2) to form the phosphatidylinositol (3,4,5)-trisphosphate (PIP3) [24]. The PIP3 acts as secondary messenger functioning as a scaffold for further downstream signaling of the BCR pathway. Among the class I PI3Ks, four isoforms may be distinguished: PI3Kα, PI3Kβ, PI3Kδ and PI3Kγ [25]. The PI3Kα, PI3Kβ and PI3Kδ are heterodimers built of a regulatory subunit, denoted as p85 (p85α, p55α, p50α, p85β, p55γ), and a catalytic subunit (p110α, p110β, p110δ). PI3Kγ is also a heterodimer comprising one of the regulatory subunits (p101 or p84) and the catalytic subunit p110γ [25]. The catalytic subunits may associate with any of the regulatory subunits. PI3Kα and PI3Kβ are ubiquitously expressed in various tissues, whereas PI3Kγ is expressed in T lymphocytes [26]. Inhibition of PI3Kγ results in impaired T lymphocyte, neutrophil and macrophage function; however, it has no effect on B lymphocytes [27,28,29].

The expression of PI3Kδ is limited mainly to the B lymphocytes and its precursors, whereas expression levels of PI3Kα and PI3Kγ are low, and PI3Kγ expression is virtually absent [30]. It was shown that PI3Kδ and PI3Kα are responsible for correct B lymphocyte development [25,30,31]. Mice deficient in PI3Kδ are characterized by diminished numbers of peritoneal (B1), follicular (B2) and marginal zone B lymphocytes, as well as defects in the humoral response and development of colitis [30,32]. Furthermore, the B lymphocytes isolated from these PI3Kδ deficient mice are characterized by a low level of protein kinase B (PKB/AKT) [32,33].

Activation of the BCR in normal B lymphocytes results in cellular differentiation, survival and antibody production [34]. Interestingly, in CLL cells, two different routes of BCR activation have been observed. The first, known as ligand-dependent, is activated via binding of an antigen or immunoglobulin M. The second is related to a tonic, constant BCR activation dependent on the activity of tyrosine kinase LYN and spleen tyrosine kinase (SYK), which ensure CLL cell survival [35]. The BCR activation signal is further propagated by activating three key enzymes i.e., phospholipase C-γ2 (PLCG2,), BTK and PI3K (Figure 1) [34]. It seems that antigen binding to BCR increases PI3Kδ activation, resulting in elevated PIP3 production and increased AKT activity, and, thus, greater B lymphocyte proliferation, differentiation, apoptosis, adhesion and migration capabilities [36]. Besides BCR activity, PI3Kγ activation may also be stimulated by the activation of membrane receptors such as chemokine CXC motif receptor 4 (CXCR4), chemokine CXC motif receptor 5 (CXCR5), CD40 protein and integrins [37,38].

PI3K activity may be diminished by the activation of antagonistic phosphatases leading to reduced levels of PIP3. The phosphatase and tensin homolog (PTEN) and SH2-containing inositol phosphatase 1 (SHIP) dephosphorylate PIP3, generating phosphatidylinositol 3-phosphate PIP1, resulting in decreased AKT activity [39]. AKT is crucial for several cellular processes as it fosters glycogen synthesis via glycogen synthase-3 (GSK3), glucose uptake via AKT substrate of 160 kDa (AS160; known as TBC1 domain family member 4 [TBC1D4]), protein synthesis via AKT substrate of 40 kDa (PRAS40), tuberous sclerosis complex 2 (TSC2) and mammalian target of rapamycin (mTOR), as well as cell proliferation and survival by regulating the activity of forkhead box (FOXO) and BCL proteins [40].

## 3. Approved PI3K Inhibitors for CLL and Lymphoid Malignancies Treatment

Three PI3K inhibitors, idelalisib, duvelisib and umbralisib, have been approved for the treatment of CLL and/or other lymphoid malignancies.

### 3.1. Idelalisib

Idelalisib (GS-1101, CAL-101, Zydelig, Gilead Sciences, Inc.; Foster City, CA, USA) was the first-in-class PI3Kδ inhibitor registered for the treatment of relapsed/refractory CLL (R/R-CLL).

#### 3.1.1. Mechanism of Action

CLL cells demonstrate far greater PI3Kδ activity than normal B lymphocytes [41]. Idelalisib treatment led to CLL cell death by inducing caspase-dependent apoptosis; this was substantial, regardless of cytogenetic factors such as 17p13 deletion or 11q22 deletion [11,41]. Furthermore, CLL cells subjected to idelalisib were characterized by reduced AKT and mitogen-activated protein kinase (MAPK) complex activity [11,37]. Idelalisib demonstrated significantly lower cytotoxic effects against non-pathological B lymphocytes, and did not show any cytotoxic effect against T lymphocytes or natural killer (NK) cells [28]. Nevertheless, reduced concentrations of IL-6, IL-10, tumor necrosis factor α (TNF-α) and interferon-γ (IFN-γ), responsible for CLL proliferation in media cultures subjected to idelalisib, were noted [41]. Despite this fact, the antigen-dependent complement cytotoxicity (ADCC) mediated by NK cells was not hindered by idelalisib therapy [41,42].

In addition to the inhibitory effects on CLL cells, idelalisib exerts its activity by reducing the synthesis of B cell activating factor (BAFF), TNF-α, and CLL3, CCL4 and CXCL13 cytokines, thus blocking the nurse-like effect of the microenvironment (Figure 1) [37,41]. Furthermore, the inhibition of CXCR4 and CXCR5 signaling pathways also results in the adhesive properties of CLL cells being lost. As a result of losing their adhesive and homing capabilities, CLL migrate to the peripheral blood, where pathological cells devoid of stimuli undergo apoptosis [38,43,44]. The described molecular mechanism is responsible for the transient increase in lymphocytosis observed in the first weeks of idelalisib treatment [45]. Considering the presented data, idelalisib acts on several key processes responsible for CLL progression, as it targets the CLL cells and disrupts the costimulatory effects of the microenvironment, while maintaining proper NK cell function.

#### 3.1.2. Clinical Efficacy

The initial phase 1 trial (#NCT00710528) was promising and allowed the maximal tolerable idelalisib dose to be established [46]. This trial included 54 heavily pretreated (median 5 lines; 2–14) patients with R/R-CLL. Idelalisib was administered daily in 50–350 mg a single dose or BID [45]. Regardless of the dose, idelalisib treatment was well tolerated with an overall response rate (ORR) of 72%; however, none of the patients achieved complete response (CR). Responses were rapid and observed within a median period of one month, with a median duration of response (DOR) of 16.2 months. The median progression free survival (PFS) was 15.8 months, while the median overall survival (OS) was not reached. Subgroup analysis showed that patients treated with 150 mg of idelalisib BID (28 patients) had a superior median PFS compared to other treatment regimens (32 vs. 7 months). On this basis, a dose of 150 mg BID was recommended for further clinical testing [45].

Idelalisib was further tested in a multicenter phase 3 randomized, placebo-controlled trial (Study 116, #NCT01539512). In patients with R/R CLL, the combination of idelalisib and rituximab (R-idelalisib) was compared to rituximab (R) monotherapy (Table 2) [47]. The study group consisted of heavily pretreated patients who had relapsed within 24 months after their last treatment, with myelosuppression preventing the use of cytotoxic agents. All participants demonstrated reduced creatinine clearance (<60 mL/min) or Cumulative Illness Rating Scale (CIRS) score greater than 6 points. In 95 patients, 17p13 deletion or *TP53* mutation was noted, and 184 patients had an unmutated IGHV status. In the event of progression, patients on rituximab monotherapy could cross over to the idelalisib arm within the extension trial (Study 117, #NCT01539291) [47]. Study 116 was stopped prematurely due to the demonstrated superiority of the idelalisib-treated arm over the control arm. After a median follow-up of 18 months, PFS was 20.3 months in the R-idelalisib arm compared to only 6.5 months of the rituximab arm [48]. Idelalisib maintained its clinical efficacy also in the high-risk patients characterized by the presence of p53 aberrations, with no significant differences in PFS being noted between these patients and those without 17p13 deletion or *TP53* mutation (median PFS 18.7 vs. 20.8 months). The overall response in the R-idelalisib arm was 85.5%, however CR was achieved only in one patient and the median DOR was 21.4 months. The analysis of the final results (Study 116 and Study 117) showed that despite the crossover, the R-idelalisib group demonstrated higher median OS than the R-placebo group (40.6 vs. 34.6 months), although only a trend toward statistical significance was observed. However, patients with p53 defects demonstrated significant benefit when treated with R-idelalisib in terms of prolonged OS compared to the control arm (28.5 vs. 14.8 months) [48]. Based on the early results of the Study 116, idelalisib was registered in 2014 in US and in Europe for the treatment of R/R CLL.

Idelalisib has been found to demonstrate clinical activity in combination with ofatumumab, the anti-CD20 monoclonal antibody [61]. The phase 3 trial Study 119 (#NCT01659021) compared ofatumumab-idelalisib to standard treatment with ofatumumab [62]. This trial corroborated the results of Study 116 by confirming the superiority of the experimental arm with regard to ORR (75.3% vs. 18.4%), lymphadenopathy reduction (93.3% vs. 4.9%) and PFS (median 16.3 vs. 8.0 months). The clinical effect was also pronounced in patients with 17p13 deletion or *TP53* mutation, with a median PFS of 13.7 vs. 5.8 months. However, no difference was observed in OS (median 20.9 vs. 19.4 months) throughout the patient group [62].

The combination of idelalisib with monoclonal anti-CD20 antibodies achieved encouraging results in trials with treatment-naïve (TN) patients [52,53]. In a phase 2 trial (#NCT01203930), 59 CLL and 5 small lymphocytic lymphoma (SLL) patients aged >65 years qualified for the R-idelalisib treatment [52]. An ORR of 97% with 19% of CR was noted; this combination was also effective in patients with 17p13 deletion or *TP53* mutation and unmutated IGHV patients with ORR 100% including 33% CR and 97% ORR with 8% CR, respectively. Median PFS and OS was not reached and 36-month PFS and OS rates were 82% and 90%, respectively. However, 19 patients (29.7%) discontinued the treatment due to AE [52]. In the phase 2 study of Lampson et al. (#NCT02135133) performed in a younger treatment-naïve CLL cohort, patients received first idelalisib 150 mg BID for 56 days and then ofatumumab was added for the next six cycles. The trial was prematurely terminated due to severe hepatoxicity, which had not been previously observed in so many patients (79% of any grade) [53]. It is noteworthy that Study 116 and Study 119 had both obsolete comparators in the control arm, as monotherapy with anti-CD20 in the relapse setting is not a standard treatment regimen.

The ASCEND trial (#NCT02970318) was a phase 3 randomized trial comparing acalabrutinib monotherapy (155 patients) to investigator choice (idelalisib + rituximab [119 patients] or bendamustine + rituximab [BR; 36 patients]); the results demonstrated the superiority of acalabrutinib, indicated by longer median PFS at a median follow-up of 16.1 months (median PFS not reached vs. 16.5 months) [51]. Estimated 12-month PFS was 88% for acalabrutinib and 68% for investigator’s choice. To note, severe adverse events (SAEs) occurred in 29% of patients treated with acalabrutinib monotherapy, 56% with R-idelalisib and 26% with BR. AEs were the reason for therapy discontinuation in 11% of patients treated with acalabrutinib and 47% treated with R-idelalisib, which could contribute to the significantly shorter PFS observed in the investigator’s choice cohort [51].

Idelalisib was also combined with bendamustine-rituximab and compared to bendamustine-rituximab in a double-blind, placebo-controlled phase 3 trial in patients with R/R CLL (#NCT01569295) [50]. Overall, 416 patients were enrolled and randomly assigned to the idelalisib and placebo groups. At a median follow-up of 14 months, median PFS was longer in the experimental arm than the placebo group (20.8 months vs. 11.1 months). However, the increased efficacy observed in the experimental arm came at the cost of increased occurrence of adverse events in the form of increased risk of infection (grade 3 and higher 39% vs. 25%), serious AEs (e.g., febrile neutropenia, pneumonia and pyrexia, 68% vs. 44%) and treatment-emergent AEs leading to death in 11% of participants, compared to 7% [50].

In the largest cohort of real-world patients published so far, idelalisib was administered in combination with rituximab in 27 patients with TN-CLL (all characterized by p53 aberrations) and 83 R/R patients (32.5% with p53 aberrant status) [60]. The median follow-up of the whole cohort was 30.2 months. The median event-free survival (EFS) was 20.3 months in the whole analyzed cohort, and 18.7 and 21.7 months in TN-CLL and RR-CLL, respectively. Median OS was not reached, but the 3-year OS rate was 56.1% [60].

#### 3.1.3. Adverse Events

PI3K possesses on-target class-specific side effects. Idelalisib currently has the best characterized AE profile, due to its long history of clinical use. The most significant non-hematological and hematological AEs of PI3K inhibitors used in CLL/SLL treatment are given below (Table 3 and Table 4).

Besides the typical profile of adverse events related to idelalisib, its use in the relapse and refractory setting led to occurrence of fatigue, diarrhea, nausea, chills and skin changes, which in most cases are graded as low-grade severity [63]. Idelalisib, like other PI3Kis, has a very specific toxicity profile which includes hepatotoxicity, diarrhea and colitis, skin changes and infections often responsible for premature drug discontinuation [63].

Diarrhea is a common adverse event characteristic for PI3Ki and, depending on clinical context, may occur in almost up to 80% of patients in any grade; however, in up to 40% cases, it may have a severe clinical course in the form of colitis (Table 3). It is more common in treatment-naïve patients and overall was shown to be responsible for drug discontinuation in approximately 10% of patients [63,64]. Two distinct forms of diarrhea may be observed during idelalisib treatment. The first type occurs normally up to eight weeks following treatment initiation and responds well to diet and antimotility agents. The second type has a late onset, following several months of idelalisib treatment, and has a potentially autoimmune character, which in some cases may lead to fatal gastrointestinal tract perforation [63,64]. Late-onset diarrhea that is unresponsive to antimotility drugs should be treated with caution and idelalisib should be withheld until the diagnostic follow-up is finished, proper treatment is administered and symptoms resolved [65,66].

Hepatotoxicity is a common AE observed during idelalisib combination therapy or monotherapy; it is characterized by elevations in alanine transaminase (ALT) and aspartate transaminase (AST) activity, which is rarely accompanied by elevation of alkaline phosphatase activity or bilirubin concentration, suggestive of hepatocellular damage. The increased ALT/AST activity is mostly noticeable in the first 8–12 weeks (median 5.3 weeks) with a plateau within week 20 of treatment in the relapse/refractory setting. The activity is often normalized following dose cessation or dose reduction of idelalisib; however, in some cases, an acute liver insufficiency based on the autoinflammatory reaction may be observed [53,63]. In treatment-naïve patients, ALT/AST elevation is observed even more often. ALT/AST elevation was noted in 79% of younger patients with CLL treated with idelalisib and ofatumumab (54% grade ≥ 3) [53]. Younger age and mutated IGHV were identified as significant risk factors for hepatotoxicity. The autoimmune nature of this finding was confirmed by the presence of lymphocytic infiltrate in liver biopsies and increased levels of the proinflammatory cytokines CCL-3 and CCL-4. Furthermore, a decrease in peripheral blood regulatory T cells was seen in patients experiencing toxicity on therapy and the elevation of transaminitis resolved following drug hold or initiation of steroids, further underlining the autoimmune nature of this adverse event [53]. Considering the growing numbers of cases with transaminitis, hepatitis B virus (HBV), hepatitis C virus (HCV), cytomegalovirus (CMV) and human immunodeficiency virus (HIV) status, this should be assessed before idelalisib treatment initiation or in case of unexpected hepatotoxicity.

Another immune related adverse event is pneumonitis, which may occur in up to 17% of cases, sometimes resulting in a fatal outcome (Table 3). So far, the largest number of cases was reported for the combination of idelalisib with entospletinib (SYK inhibitor) which led to development of severe pneumonia/pneumonitis in 11 (17%) patients, of which 5 (8%) were in need of mechanical ventilation and 2 were fatal due to progressive respiratory insufficiency [67]. Pneumonitis should be suspected in patients with acute cough accompanied by dyspnea and fever after several months (median eight months till onset) of idelalisib treatment. The radiological findings report diffuse ground-glass opacities, mostly bilateral with possible pleural effusions which may initially be hard to distinguish from an infectious etiology. Considering that increased number of Pneumocystis jiroveci (PJP) pneumonia and CMV reactivations are a potential ethological factor of pneumonia (especially in idelalisib combination treatment), a careful diagnostic follow-up is necessary [63,68,69,70].

Neutropenia is also a common AE during the first weeks of idelalisib treatment; anemia and thrombocytopenia are less often observed. Neutropenia may occur in about half of the patients, and in approximately 20% of those with grade 3–4 (Table 4). Neutropenia is associated with an increased rate of infection, including opportunistic ones (PJP and CMV); therefore, granulocyte colony-stimulating factor (G-CSF) support, prophylactic treatment with trimethoprim-sulfamethoxazole (TMP-SMX) and CMV status assessment with CMV DNA monitoring is recommend during and after idelalisib treatment [63,71]. An interim analysis of six clinical trials involving idelalisib combinations in CLL and FL revealed an increased risk of death in arms treated with PI3Ki combinations compared to control arms (7.4% vs. 3.5%) leading to temporary hold of clinical trials in March 2016 [71]. Beside the above-mentioned prophylactic work up, acyclovir prophylaxis is also recommended due to potential severe skin infections and Varicella zoster infections.

Similar adverse event profiles have been obtained from clinical trials and real-world patients, and no new safety issues have been observed. In the largest observational study published so far, including 110 patients (27 TN-CLL and 83 RR-CLL), lower respiratory tract infection/pneumonia were reported in 34.5% (grade ≥ 3, 19.1%), diarrhea in 30.9% (grade ≥ 3, 6.4%), and colitis in 9.1% (grade ≥ 3, 5.5%) [60]. In total, idelalisib was discontinued in 87.3% patients. However, while more patients discontinued their treatment due to adverse events in the front-line setting compared to relapse/refractory patients (63.0% vs. 44.6%), the greater discontinuation was observed due to progressive disease in the relapse/refractory patients than in the patients with TN-CLL (20.5% vs. 3.7%). This confirms that treatment-naïve patients tend to demonstrate poorer idelalisib tolerance [60].

The combination of idelalisib with compounds other than monoclonal antibodies does not result in substantially increased efficacy and leads to serious adverse events in some cases [67,72,73]. The combination of idelalisib, lenalidomide and rituximab was poorly tolerated and led to premature termination of two clinical trials due to severe life threatening serious adverse events. Smith et al. reported the occurrence of grade ≥ 3 transaminitis, rash, hypotonia, pneumonia and sepsis-like syndrome in eight patients [72], while Cheah et al. noted the presence of grade ≥ 3 transaminitis in six out of seven patients, with two deaths: one from acute liver failure and the other due to progressive respiratory failure in the course of Gram-positive sepsis [73].

Considering the accumulated data and the type of adverse events, the on-target blockade of the PI3Kδ by idelalisib in normal B lymphocytes and T regulatory cells (Tregs) modifies the immunotolerance increasing the cytotoxic activity of CD8+ cells and loss of immunological tolerance [71,74]. This idea is corroborated by histopathological examination presenting CD8+ lymphocytic infiltrates in biopsies of the liver [53], mucous membrane of the colon [65,66,75] and lung [63]. PI3Kδ activity is necessary for Treg function, and the genetic loss of the kinase or its blockade by idelalisib is toxic and leads to a loss of immune tolerance [76,77,78,79].

One possible way to reduce on-target adverse events is to adjust the idelalisib dosing to effectively block neoplastic cells while maintaining proper immunoregulatory Treg function. It was shown that human Tregs recover after two weeks of withdrawal of cytotoxic agents [80]. Unlike ibrutinib, idelalisib withholding does not lead to rapid disease recurrence and in some patients, the response to treatment is sustained despite idelalisib withdrawal [75]. One study found noticeable clinical benefit that, in patients with CLL after drug withdrawal following idelalisib therapy longer than six months, interruption with dose reduction did not worsen clinical outcomes. However, prolonged off-therapy time may diminish the clinical benefit of treatment interruption [81]. Therefore, intermittent dosing of idelalisib, which would allow Treg recovery during the off-drug period, is a plausible option, however it has not so far been tested in a clinical trial setting.

### 3.2. Duvelisib

Duvelisib (IPI-145, INK1197, Copiktra, Secura Bio, Inc.; Las Vegas, NV, USA) is a first-in-class, highly selective and potent dual inhibitor of PI3Kδγ. In contrast to PI3Kδ, which is expressed mostly on leukocytes, PI3Kγ is mainly expressed on T cells and on myeloid cells. PI3Kγ inhibition therefore is aimed at reducing the production of cytokines promoting leukemic cell survival. In preclinical studies, duvelisib exposure resulted in direct cytotoxicity to leukemic B cells and reduced production of pro-survival cytokines [82,83,84]. In animal models, dual PI3Kδγ inhibition with duvelisib was found to be more active than blocking either isoform alone [85,86,87].

#### 3.2.1. Clinical Efficacy

The clinical development of duvelisib started from a phase I open label, dose-escalation and cohort expansion study, IPI-145-02, with 210 patients with advanced hematologic malignancies including 73 (35%) patients with CLL/SLL (#NCT01476657) [54]. Both R/R CLL/SLL and TN-CLL patients were enrolled; however, all patients with TN-CLL required ≥65 years of age and/or 17p deletion and/or *TP53* mutation. In the dose-escalation phase, the maximum tolerated dose was 75 mg twice daily; however, doses as high as 25 mg BID caused inhibition of CLL proliferation and decrease of the cytokines and chemokines stimulating B cell proliferation. Based on the combined efficacy, safety, pharmacokinetics and pharmacodynamic data, a dose of 25 mg twice daily was selected for subsequent phase 2 and 3 studies [54]. The ORR in patients with R/R CLL/SLL and TN-CLL was 56% and 83%, respectively, and median PFS was 15.7 months in R/R CLL/SLL patients; OS was not reached in either cohort (Table 2) [88]. Importantly, duvelisib was efficient in patients with 17p deletion/*TP53* mutation (ORR 46%) as well in those with unmutated IGHV (ORR 51%).

In the phase 2 global open-label, DYNAMO trial (#NCT01476657), 129 patients with relapsed refractory indolent non-Hodgkin lymphoma (iNHL) including SLL (28 patients), follicular lymphoma (FL) and marginal zone lymphoma (MZL), refractory to rituximab and either chemotherapy or radioimmunotherapy were enrolled. The ORR was 47% in all iNHL patients and 68% in the cohort of SLL patients, with all the responses being classified as partial response (PR) (Table 2) [55].

Duvelisib was approved by FDA in 2018 and by EMA in 2021 for the treatment of R/R-CLL/SLL patients after at least two prior therapies based on the results of a global phase, multicenter, randomized open-label phase 3 DUO trial (#NCT02004522) [56]. Three hundred and nineteen CLL/SLL patients that progressed after at least one prior therapy and were not previously treated with BTK or PI3Kδ inhibitors were randomized to receive duvelisib or ofatumumab. Patients on either treatment arm with radiographically confirmed disease progression were given the option to cross over to receive the opposite therapy in a separate extension study. Del(17p)/*TP53* mutation and unmutated IGHV status were found in 31% and 59% of patients treated with duvelisib, respectively, and in 33% and 73% of the patients treated with ofatumumab. As assessed by a blinded independent review committee (IRC), the PFS was significantly longer and ORR was higher in patients treated with duvelisib (13.3 vs. 9.9 months and 74 vs. 45%, respectively) regardless of del(17p)/TP53 status (Table 2). In the duvelisib arm, 72.5% of the patients achieved a PR. Ninety patients with CLL progression after ofatumumab crossed over to duvelisib in the DUO trial and 60 (77%) achieved a response with PFS of 15.7 months. Similar outcomes were noted in patients with del(17p)/TP53 mutation (ORR, 77%; PFS, 14.7 months) and in patients primary refractory to ofatumumab (ORR, 73%) [89].

Duvelisib was also evaluated in combination with FCR immunochemotherapy. However, the results of phase a 1b/2 study with the 3-year PFS of 73% did not show the superiority of duvelisib and FCR regimen over a historical group treated in the first line with FCR (#NCT02158091) [90]. In the R/R-CLL setting, duvelisib was combined with rituximab (11 patients) or bendamustine and rituximab (six patients) in a phase 1 study (#NCT01871675) [91]. The ORR was found to be 89% for patients treated with duvelisib and rituximab and 75% for those treated with duvelisib and BR; these values were not higher than those obtained for duvelisib monotherapy in the DUO study. However, it was not possible to define the efficacy due to the small patient groups. The results of ongoing studies on duvelisib efficacy and safety in non-chemo combination regimens such as venetoclax might justify the use of duvelisib in time-limited regimens. Duvelisib is also undergoing evaluation in intermittent dosing, to reduce toxicity, or in combination with nivolumab (Table 5).

#### 3.2.2. Adverse Events

Duvelisib toxicities (grade 3 or higher) of clinical significance in patients with CLL are typical of PI3Ki, and include diarrhea, elevated AST or ALT, colitis, pneumonitis and infectious complications.

In the IPI 145-02 trial of duvelisib, most events were grade 1 or 2, and the most common all grade toxicities were diarrhea (42%) fatigue (40.5%), ALT increase (38.5%), elevated AST (37.6%), pyrexia (35.2%), nausea (32%) and cough (31.4%) [54]. The most common grade ≥3 toxicities were ALT increase (19.5%), AST increase (15.3%) and diarrhea (11.4%) (Table 3). Hematological complications were relatively common, with neutropenia (all grades) observed in 38.6% of patients and grade ≥ 3 in 20% of patients, anemia in 24.8% and 14.3%; thrombocytopenia in 23.3% and 14.3% of patients, respectively (Table 4). Infections, regardless of grade, were reported in 61% of patients, with upper respiratory tract infections in 16.2% and pneumonia in 13% (10% grade ≥ 3). PJP pneumonia was noted in three patients (1%); no patients received prophylaxis, and CMV infection occurred in two patients. Severe events of colitis and pneumonitis were reported in 6% and 4% of patients, respectively. The incidence of AE leading to treatment discontinuation was 36% [54].

In the DYNAMO trial, the median duration of treatment exposure was 6.7 months (range, 0.4 to 45.5 months). The most frequent any-grade AE were diarrhea (48.8%), nausea (29.5%) and elevated ALT and AST (14% and 10.1%, respectively). Colitis and pneumonitis were both reported in 7.8% of patients. Grade ≥ 3 events were infrequent and included diarrhea (14.7%), colitis and pneumonitis (both 5.4%), elevated ALT (5.4%) and AST (3.1%) (Table 3). Hematological complications were relatively common with neutropenia noted in 28.7% of patients (grade ≥ 3, 24.8%), anemia in 26.4% (grade ≥ 3, 14.7%) of patients and thrombocytopenia in 18.6% (grade ≥ 3, 11.6%), respectively (Table 4) [55]. Three patients experienced serious opportunistic infections: bronchopulmonary aspergillosis, cytomegaloviral pneumonia, and PJP in a patient with prescribed prophylaxis on day 1; 31% of patients discontinued duvelisib due to treatment-emergent adverse events (TEAE).

In the DUO trial, median treatment exposure for duvelisib was 50 weeks. The most common hematological AEs were neutropenia (33%), anemia (23%) and thrombocytopenia (15%). The most common non-hematological AEs in the duvelisib-treated group were diarrhea (51%), pyrexia (29%), nausea (23%) and cough (21%). Colitis was reported in 13% of patients. Median time to first event of diarrhea or colitis was four and seven months, respectively (Table 4). The most common AE ≥ 3 were neutropenia (30%), diarrhea (15%), pneumonia (14%) and anemia (13%). Severe immune-related toxicities were colitis (12%) and pneumonitis, ALT or AST increase (3% each). None of the events were fatal. Infectious AE were reported in 69% of patients treated with duvelisib, including 18% of pneumonia and 16% of URTI (upper respiratory tract infection). PJP pneumonia occurred in three patients treated with duvelisib that were not receiving prophylaxis despite being required per protocol [56].

The combination of duvelisib with fludarabine, cyclophosphamide and rituximab (FCR) was associated with high toxicity. Grade ≥ 3 neutropenia was reported in 62% of patients, febrile neutropenia in 22%, lymphopenia in 66%, ALT increase in 29% and AST increase in 24% [89]. Combination of duvelisib with rituximab or BR did not increase the toxicities of individual agents. Neutropoenia grade ≥ 3 was noted in 41% of patients, diarrhea in 13.1%, ALT increase in 6.5%, AST increase in 2.2% and pneumonia in 6.5% of patients [91].

Due to the rare but fatal events observed in clinical studies, duvelisib carries black-box warnings for serious infections, severe diarrhea or colitis, serious cutaneous reactions and pneumonitis. Patients on duvelisib need careful monitoring and proper management of adverse events. Prophylaxis for PJP and CMV is mandatory throughout the duration of duvelisib treatment, as is monthly CMV viral load monitoring. Duvelisib should be withheld until infection resolution and discontinued if PJP is confirmed. In the case of autoimmune complications (diarrhea, colitis, pneumonitis) duvelisib should be withheld and steroid therapy initiated; in cases of grade 3 or higher, the discontinuation should be permanent. The treatment should be also withheld in the case of grade 4 hematological toxicities. It is important to consider drug interactions, since strong CYP3A4 inducers may decrease drug efficacy. When given together with strong or moderate CYP3A inhibitors, duvelisib should be dosed at 15 mg twice daily.

### 3.3. Umbralisib

Umbralisib (Ukoniq, TG Therapeutics; Morrisville, NC, USA) is a next-generation highly selective PI3Kδ inhibitor that additionally inhibits casein kinase-1ε (CK1ε), which is involved in the translation of c-Myc oncogene and the regulation of the Wnt5a pathway [14,92]. It has a different chemical structure and safety profile to other PI3Kδ inhibitors. CK1ε inhibition blocks the effects of PI3K inhibition, thus sparing Treg numbers and function, and reducing the incidence of autoimmune complications [93,94].

#### 3.3.1. Clinical Efficacy

The safety and preliminary efficacy of single-agent umbralisib was evaluated in an open-label, dose escalation phase 1 study (#NCT01767766) [57]. In a cohort of 90 patients with R/R CLL, B-, and T-NHL, or Hodgkin lymphoma, umbralisib was administered in a dose escalating from 50 to 1800 mg once-daily; the recommended dose for phase 2 study was determined to be 800 mg once daily. Umbralisib monotherapy resulted in in high ORR of 85% in relapsed CLL patients, including 7 PR with lymphocytosis (Table 2). Similar efficacies were observed between patients with CLL with high-risk and normal cytogenetics.

A subsequent phase 2 study also evaluated umbralisib in patients with CLL who were intolerant to prior BTKis (n = 44) or PI3Kδis (n = 7). Median PFS was 23.5 months, and 58% of patients were on umbralisib for longer than prior BCRi. The data confirmed that switching from BCRi or alternate PI3Kito umbralisib is safe and can result in favorable responses [95].

Its favorable toxicity profile resulted in umbralisib being tested in different combinations. In a multicenter phase 1b study, continuous umbralisib and ibrutinib treatment was administered until progression or unacceptable toxicity with the aim to determine the maximum tolerated dose, safety and efficacy. The study included 21 patients with CLL and 21 patients with MCL. The maximal tolerated dose (MTD) of umbralisib was not reached. The treatment resulted in 90% rate of ORR including 29% CR in 21 patients with relapsed CLL. The PFS was 90% at a two-year follow-up [96].

The most thoroughly studied combination treatment is the pairing of umbralisib with ublituximab, a novel type-1 glycoengineered monoclonal anti-CD20 antibody, and enhanced ADCC (antibody-dependent cellular cytotoxicity); this is known as the U2 protocol. In a phase 1/1b escalation study (#NCT02006485) conducted in the patients with NHL and CLL, the final recommended doses were determined as 800 mg of oral umbralisib daily and 900 mg of IV ublituximab. In 22 CLL/SLL patients, ORR was 67%, (40% with prior BTKi), DOR was 25.89 months and PFS was 25.57 months (Table 2). No new safety issues were observed compared to umbralisib monotherapy [58].

The efficacy and safety of U2 were then compared with those of obinutuzumab and chlorambucil (O + Chl) in a multicenter, phase 3 UNITY-CLL trial (#NCT02612311) that enrolled patients with both TN-CLL and R/R-CLL [81]. In general, 421 patients were randomized to U2 and the median treatment duration was 23 months. At the median follow-up of 36.2 months, PFS was significantly longer in patients treated with the U2 regimen compared to O + Chl (median 31.9 months vs. 17.9 months; *p* < 0.0001) (Table 2). IRC-assessed ORR was higher with U2 (83.3%) vs. O + Chl (68.7%; *p* < 0.001). In patients previously treated with ibrutinib, ORR was 57% for U2 compared to 25% for O + Chl. A further analysis of the UNITY-CLL study showed that patients with BTKi risk factors defined by a preexisting comorbidities profile attained favorable clinical benefit with U2, indicating it to be an effective and safe alternative to BTKi [97].

Recently, Roeker et al. presented the results of the first minimal residual disease (MRD)-driven study based on the combination of BTKi, PI3Ki and anti-CD20 monoclonal antibodies. In this phase 2 study, umbralisb and ublituximab (U2) were added to ibrutinib for patients treated with ibrutinib in any line of therapy for a minimum of six months with detectable MRD in peripheral blood. Treatment was terminated after undetectable MRD (uMRD) was achieved and confirmed four weeks later, but was continued for up to 24 cycles in patients with sustained MRD. Undetectable MRD was achieved in 71% of patients with a median time to the first uMRD result of five months. Ibrutinib with U2 was well tolerated; grade 3/4 AEs of clinical interest included ALT/AST increase (4%), diarrhea (4%) and hypertension (8%). These findings showed that the combination with PI3Ki was effective and well tolerated based on an MRD-driven time-limited approach [98]. Several protocols with umbralisib, most with ublituximab, are under evaluation in patients with CLL, including combinations with acalabrutinib, ibrutinib or venetoclax (Table 5).

#### 3.3.2. Adverse Events

Due to its better selectivity for δ isoforms, umbralisib is characterized by a favorable safety profile with much less immune-mediated toxicities than the other PI3Kδ inhibitors (idelalisib, duvelisib). It is possible that autoimmunity is minimized by augmentation of regulatory T-cell activity through CK1ε inhibition.

In the phase 1 study, the most common AEs associated with umbralisib monotherapy were diarrhea (43%), nausea (42%) and fatigue (31%). In most cases, diarrhea occurred early (median 50 days since the start of therapy) and resolved without intervention. The most common grade 3 or 4 events were neutropenia (13%), anemia (9%) and thrombocytopenia (6%); ALT or AST increase was observed in 2% and 3%, respectively, diarrhea in 3% and colitis in 2% (Table 3 and Table 4). Discontinuation of umbralisib due to AE was infrequent and occurred only in 7% of patients [57].

The combination of umbralisib and ublituximab was well tolerated when both drugs were used in standard doses. The rates of all grade infection and diarrhea were higher comparing to each drug monotherapy; however, most events were low grade. The incidence of immune-mediated complications was low (Table 4). The most common AEs of any grade in patients with CLL were nausea (82%), diarrhea (64%), fatigue (50%) and neutropenia (55%). Diarrhea had a median time to onset of 21 days and resolved in a median of seven days. Grade 3 AE were uncommon there were no grade 3 or higher diarrhea or transaminase elevation [58]. The most frequent grade 3 AE was neutropenia (50%). The incidence of opportunistic infections was low with no events of PJP or CMV observed, though the follow-up was relatively short (median 7.4 months).

In the UNITY CLL, median follow-up was 36.2 months and the median U2 treatment duration was 23 months. Grade 3/4 AEs of interest in patients treated with U2 included neutropenia (30.6%), diarrhea (12.1%), elevated AST/ALTs (8.3%), colitis and pneumonitis (2.9% each). AEs led to treatment discontinuation in 34 patients (16.5%) treated with U2 regimen [59,85].

In February 2021, the Food and Drug Administration (FDA) granted accelerated approval to umbralisib for patients with R/R marginal zone lymphoma (MZL) and patients with R/R follicular lymphoma (FL). However, umbralisib is not yet approved for CLL.

## 4. Novel PI3K Inhibitors in Clinical Studies

Since the PI3K pathway has been implicated in a number of cancers, many PI3Ki agents are in preclinical and clinical development in the treatment of solid tumors and hematological malignancies. Most of the studies with PI3K pan-inhibitors are conducted in solid tumors, with the majority in early preclinical and clinical stages. Selective and dual PI3K inhibitors are under development in both solid tumors and in B and T cell lymphoma and CLL. Novel agents in early stages of development in patients with CLL are zandelisib, parsaclisib, BGB-10188, tenalisib, ACP-319, HMPL-689, SHC014748M and TQ-B3525 (Table 1).

### 4.1. Copanlisib

Copanlisib (BAY 80-6946, Aliqopa™, Bayer, *Leverkusen*, Germany) is an inhibitor of multiple PI3K isoforms (p110α, p110β,p110 γ and p110δ) [13]. The drug is approved by the FDA and European Medicines Agency (EMA) for the treatment of patients with R/R-FL. Copanlisib is being studied in combination with ibrutinib in patients with R/R-CLL. The drug is approved by FDA for R/R-FL after at least two prior therapies. When used in monotherapy in relapsed patients with CLL, response rates were 40% to 75% [99,100]. A phase 2 study evaluating copanlisib combined with ibrutinib in R/R-CLL is ongoing (#NCT04685915).

### 4.2. Zandelisib

Zandelisib (PWT143, ME-401, Mei Pharma, San Diego, CA, USA) is an investigational selective PI3Kδ inhibitor with a molecular structure distinct from other PI3Kis and longer PI3Kδ occupancy [15,101]. Zandelisib demonstrated target binding in vitro longer than five hours. It has prolonged target inhibition and a high volume of distribution, indicating high tissue exposure [102,103]. The drug is in clinical development for the treatment of patients with B cell lymphoid malignancies, including RR-CLL and B cell NHL. In a phase 1 study of zandelisib, used alone or in combination with rituximab, durable objective responses were observed in R/R indolent B cell malignancies (FL, CLL/SLL, MZL and diffuse large B-cell lymphoma [DLBCL]; #NCT02914938) [104]. The overall response rate was 83%, including 89% in CLL/SLL (100% in monotherapy and 83% in combination group), with median DOR not reached. The drug was well tolerated without safety differences between the monotherapy and rituximab combination groups. Grade 3 and higher AEs were reported in seven of 57 patients, including diarrhea (3.5%), colitis (3.5%), rash (2%), alanine transferase increased (2%), and pneumonitis (2%). A randomized, open-label, controlled multicenter phase 3 study (COASTAL) to investigate the safety and efficacy of zandelisib in combination with rituximab versus standard immunochemotherapy in patients with RR-iNHL is ongoing (#NCT04745832) [105]. A phase 2 trial with zandelisib combined with rituximab and venetoclax in patients with RR-CLL (CORAL) has also been initiated (#NCT05209308).

### 4.3. Parsaclisib

Parsaclisib (INCB50465, IBI-376, Incyte, Wilmington, DE, USA) is a next next-generation, highly selective PI3Kδ inhibitor with approximately 20,000-fold selectivity for PI3Kδ over other isoforms (PI3Kα, PI3Kβ, and PI3Kγ). In the first-in-human phase 1/2 CITADEL-101 study, parsaclisib demonstrated antitumor activity in relapsed or refractory B-cell NHL (#NCT02018861) [106]. The drug was studied alone or in combination with itacitinib (Janus kinase 1 inhibitor) or chemotherapy with rituximab, ifosfamide, carboplatin and etoposide in patients with R/R B cell malignancies (#NCT02018861) [106]. Parsaclisib has demonstrated antitumor efficacy and an acceptable safety profile. It was also evaluated in monotherapy in an open-label, multicenter, phase 2 study in patients with R/R-DLBCL (CITADEL-202 study, #NCT02998476) [107]. The drug was well tolerated; however, the ORR was only 25.5% (eight complete metabolic responses and six partial metabolic responses) and the median DOR was 6.2 months. Parsaclisib is currently under evaluation in combination with the anti CD19 monoclonal antibody tafasitamab in patients with R/R-CLL and R/R-NHL as part of a phase 1/2 study (topMIND, #NCT04809467).

### 4.4. BGB-10188

BGB-10188 (BeiGene; Beijing, China) is a highly selective inhibitor of PI3Kδ, which does not demonstrate significant inhibition of other 376 protein kinases and 17 lipid kinases [18]. This agent shows more than 3000-fold greater selectivity over PI3Kα, PI3Kβ and PI3Kγ. In preclinical studies, it demonstrated an improved safety profile in comparison with other PI3Kis. A phase 1/2 study of BGB-10188, as monotherapy and in combination with zanubrutinib and tislelizumab, in patients with CLL and other B cell NHLs, is ongoing (#NCT04282018).

### 4.5. Tenalisib

Tenalisib (GDC-0032, RP6530, Rhizen Pharmaceuticals SA; Basel, Switzerland) is a novel, orally available, dual PI3K δ/γ inhibitor with several-fold selectivity over PI3Kα and PI3Kβ isoforms. The drug induced apoptosis and showed anti-proliferative activity in B and T lymphoma cell lines [108]. In addition, it also demonstrated antitumor activity in a T cell leukemia xenograft model in a mouse model and in patient-derived primary cutaneous T cell lymphoma (CTCL) cells [108]. Recently, it demonstrated an acceptable safety profile and promising clinical activity in patients with R/R T cell lymphoma [109]. A phase 2 study investigating the efficacy and safety of tenalisib in patients with R/R-CLL after at least one prior therapy has been also initiated (#NCT04204057).

### 4.6. ACP-319

ACP-319 (AMG 319, Acerta Pharma BV (Redwood City, CA, USA)/AstraZeneca (Cambridge, UK) is a novel selective PI3Kδ inhibitor with promising results in early preclinical and clinical studies in CLL and other lymphomas [100,101,102]. In the first human study of AMG 319, the safety, tolerability and pharmacokinetics of the drug were evaluated in 28 heavily pretreated patients with R/R-CLL (n = 25) and R/R-NHL [110]. Among 24 evaluable patients with CLL and 3 with NHL, all had >50% lymph node (LN) reduction as a best response. Responses were present in all high-risk cytogenetic subgroups of CLL. Phase 1 study of ACP-319 combined with acalabrutinib in R/R CLL is ongoing (#NCT02157324).

### 4.7. HMPL-689

HMPL-689 (Hutchison China MediTech; Hong Kong, China) is a novel class I PI3Kδ inhibitor [111]. In a dose-escalation phase I study in 56 patients with R/R-NHL, including CLL, it was found to have good safety profile and promising efficacy [112]. Objective response was noted in 27 of 51 (52.9%) evaluated patents (including six CR, of which two were CLL/SLL). HMPL-689 was well tolerated, with a manageable safety profile. The most common grade 3 or higher non-hematologic treatment emergent AEs were pneumonia and hypertension Similar results were also reported from phase1b study [113]. A total of 75 patients with CLL and other NHL had received at least one dose of HMPL-689. The ORR was 51.7% and the median time to response (TTR) was 1.9 months. The most common treatment-emergent AEs were neutropenia, ALT and AST increase, leukopenia, hypertriglyceridemia, pneumonia and upper respiratory tract infection. The most common grade 3 or higher AEs were neutropenia, pneumonia and rash.

### 4.8. SHC014748M

SHC014748M (Nanjing Sanhome Pharmaceutical; Yunliang, China) is an inhibitor of PI3K with high selectivity to the δ isoform. SHC014748M demonstrated promising preclinical antitumor activity in B-cell NHL and CLL [22]. The agent inhibited cell proliferation both in B cell lymphoma cell lines and in primary CLL cells. In addition, SHC014748M induced apoptosis of lymphoma cell line in a concentration-dependent manner. It also demonstrated antineoplastic activity in SU-DHL-6 xenograft model. Phase 1 study SHC014748M in patients with CLL and other indolent B-cell hematologic malignancies has been registered (#NCT03588598).

### 4.9. TQ-B3525

*TQ-B3525* (Chia Tai Tianqing Pharmaceutical Group; Lianyungang, China) is a selective PI3Kα/δ inhibitor being investigated in B and T cell lymphoid malignancies (#NCT04808570, #NCT04615468). In preclinical studies, *TQ-B3525* demonstrated 41-fold higher activity against PI3Kα and 138-fold higher against PI3Kδ compared to buparlisib [23]. In a phase 1 study performed in 27 patients with R/R lymphoma and 13 patients with advanced solid tumor, the most common AEs of all grades were hyperglycemia (65.0%), glycosylated hemoglobin increased (35.0%) and diarrhea (32.5%). DLT was grade 3 hyperglycemia, observed in three patients. Among 23 lymphoma patients, ORR was 60.9% and median PFS was not reached. *TQ-B3525* in patients with R/R CLL/SLL is ongoing (#NCT04808570).

## 5. Resistance Mechanism to PI3K Inhibitors in Chronic Lymphocytic Leukemia

Although PI3K inhibitors are often used for the treatment of solid cancers, some potential mechanisms of resistance have been identified. The overall effectiveness of PI3K inhibitor compounds has been greatly limited by the reactivation of the cross-over pathways by compensatory mechanisms [40]. Cancer cells were shown to activate downstream pathways or induce alternative pathways, such as RAF-MAPK and NOTCH, as well as c-MYC activation [114,115]. In addition, the loss of PTEN expression or altered expression/activity of various kinases, such as RSK3/4, PIM, FOXM1, PDK1-SGK1, SGK3 and CD4/6, may also contribute to the development of PI3K resistance [40].

PI3K resistance mechanisms in CLL are not well defined and only few studies have addressed this problem. In contrast to treatment with BTK inhibitors, it appears that mutations in PI3K and related-pathway genes are rarely responsible for therapy failure [116,117]. Whole-exome sequencing was performed in 13 patients with CLL showing progression following initial response, who were enrolled in phase 3 studies (study 116, study 117 and study 119). Twelve of these patients had an unmutated IGHV status and eight had p53-pathway aberrations. Interestingly, no recurrent progression-associated mutations were identified in more than one patient, and no progression-associated mutations were identified in the PI3K signaling pathway or in any other related pathway [118].

It is possible that the mechanism of PI3K inhibitor resistance observed in CLL may be based on the activation of alternative pathways via occurrence of novel mutations of genes of cross-signaling pathways or their reactive overactivation, as is the case in solid cancers, although data are scarce in this regard [40]. Longitudinal WES of cells from patients with RR-CLL found activating mutations in MAP2K1, BRAF and KRAS genes in 60% of patients [119]. In non-responder CLL cells with and without these mutations, PI3Kδ inhibition failed to inhibit ERK phosphorylation (pERK); however, treatment with a MEK inhibitor allowed for successful ERK inhibition. Overexpression of MAP2K1 mutants in vitro led to increased basal and inducible pERK and idelalisib resistance, demonstrating that MAPK/ERK activation may play a role in PI3Kδ treatment failure [119]. These results are corroborated by a mouse model of CLL in which resistance mechanism to PI3Kδ inhibition was achieved by serial transplantation of induced tumors [120]. Resistance to the PI3Kδ inhibitor was mediated by functional activation of insulin-like growth factor 1 receptor (IGF1R), resulting in increased MAPK signaling. The IGF1R upregulation in the resistant tumors was mediated by functional activation and enhanced nuclear localization of forkhead box protein O1 transcription factors (FOXO1) and glycogen synthase kinase 3β (GSK3β). The overexpression of IGF1R in vitro confirmed its role in this resistance mechanism [120]. The existing body of literature demonstrates that selective inhibition of potentially PI3Kδ resistance-inducing pathways may be a good strategy for maintaining therapeutic efficacy.

## 6. Conclusions

PI3Kis are a heterogenous group of small molecule agents used to treat a range of cancer types. Blockade of the target enzyme results in various biological consequences depending on whether the agent is a pan-inhibitor, isoform-specific inhibitor (with different selectivity) or dual inhibitor. Pan-inhibitors have mainly been studied in various solid tumors, both as monotherapy and in combination with chemotherapy, with only copanlisib being approved by the FDA in lymphoid malignancies (follicular lymphoma). Idelalisib, copanlisib and recently umbralisib are also approved for this indication, with the latter also being approved in MZL. In CLL, two PI3K inhibitors, idelalisib and duvelisib, have been approved; however, despite demonstrating similar efficacy, they are used much less often than BTKi due to their poorer toxicity profile, including mainly autoimmune complications such as colitis, pneumonitis and increased susceptibility for infections.

However, due to the poor prognosis associated with the failure of the approved small-molecule agents, new options are still needed. In response, novel agents with a distinct safety profile, such as umbralisib, are under development in CLL in different combinations, with MRD-driven, time-limited approaches also being used. Furthermore, a number of PI3Ki agents are in preclinical and early clinical development, and one can expect that their role in the treatment of patients with CLL to grow.

## Figures and Tables

**Figure 1 cancers-14-01571-f001:**
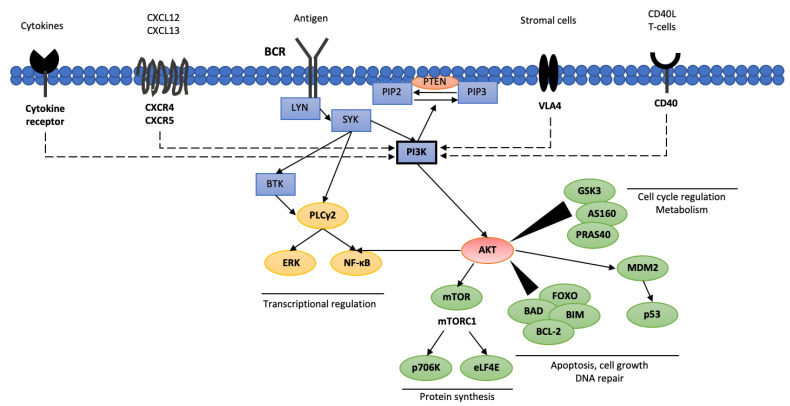
Phosphoinositide 3-kinases interacting pathways in CLL cells. AKT—protein kinase B (PKB); AS160—Akt substrate of 160 kDa; BCL-2—B cell lymphoma 2; BIM—bcl-2-interacting mediator of cell death; BAD—BCL2 associated agonist of cell death; BCR—B cell receptor; BTK—Bruton’s tyrosine kinase; CXCL12—C-X-C motif chemokine 12; CXCL13—C-X-C motif chemokine 13; eLF4E—eukaryotic translation initiation factor 4E; ERK—extracellular signal-regulated kinase; FOXO—forkhead box protein; GSK3—glycogen synthase kinase 3; LYN—tyrosine-protein kinase Lyn; MDM2—mouse double minute 2 homolog; NF-κB—nuclear factor kappa-light-chain-enhancer of activated B cells; p706K—p706 kinase; PI3K—phosphoinositide 3-kinases; PIP2—phosphatidylinositol 4,5-bisphosphate; PIP3—phosphatidylinositol 3,4,5-triphosphate; SYK—spleen tyrosine kinase; PLCγ2—phospholipase C γ2; PTEN—phosphatase and tensin homolog; mTOR—mammalian target of rapamycin; PRAS40—proline-rich Akt substrate of 40 kDa; VLA4—very late antigen 4.

**Table 1 cancers-14-01571-t001:** Phosphoinositide 3-kinase inhibitors (PI3Ki) approved or potentially useful for chronic lymphocytic leukemia.

Agent	Target	IC_50_, nM	Dose	FDA/EMA Approval
Idelalisib/CAL-101, GS-1101 [11]	PI3Kδ	2.5	150 BID, po	CLL, SLL FL
Duvelisib/IPI-145, INK1197 [12]	PI3Kδ/PI3Kγ	2.5/27	25 mg, po, BID	CLL FL
Copanlisib/BAY 80-6946 [13]	PI3Kα/PI3Kδ	0.5/0.7	60 mg, iv, QD	FL/NA
Umbralisib/TGR-1202, RP5264 [14]	PI3Kγ and casein kinase-1ε [CK1ε])	22.3	800 mg, po, QD	FL MZL/NA
Zandelisib/ME-401, PWT143 [15]	PI3Kδ	5	60 mg, po, QD	NA
Parsaclisib/INCB050465 [16,17]	PI3Kδ	1.1	20 mg, po, QD	NA
BGB-10188 [18]	PI3Kδ	1.7–16	NK, po, QD	NA
Tenalisib/RP6530 [19]	PI3Kδ/PI3Kγ	24/33	800 mg, po, BID	NA
ACP-319, AMG 319 [20]	PI3Kδ	18	NK	NA
HMPL-689 [21]	PI3Kδ	0.8	30 mg, po, QD	NA
SHC014748M [22]	PI3Kδ	0.77	NK, po, QD	NA
*TQ-B3525* [23]	PI3Kα/PI3Kδ	4.2	20 mg, po, QD	NA

CLL—chronic lymphocytic leukemia; FL—follicular lymphoma; MZL—marginal zone lymphoma; po—per os; iv—intravenous; NA—not approved; NK—not known; SLL—small lymphocytic lymphoma.

**Table 2 cancers-14-01571-t002:** Selected clinical studies with PI3Ki in patients with CLL/SLL.

Study Regimen	Phase/Name	Population	Number of Patients (Patients with CLL/SLL)	Median Age (Years)	Primary Endpoint	ORR (%) *	CR *	Median PFS *	Median OS *	Reference
Idelalisib + rituximab vs. rituximab	3/study 116	RR-CLL	220	71	PFS	85.5	1 patient (<1%)	20.3 months	40.6 months	[48]
Idelalisib + ofatumumab vs. ofatumumab (Study 119)	3/study 119	RR-CLL	261	67	PFS	75.3	1 patient (<1%)	16.3 months	20.9 months	[49]
Idelalisib + BR vs. placebo + BR	3	RR-CLL	416	62	PFS	70.0	1.4	20.8	NR	[50]
Acalabrutinib vs. BR/idelalisib	3/ ASCEND	RR-CLL	398	68	PFS	not reported separately	not reported separately	16.3 months for investigator choice	NR	[51]
Idelalisib + rituximab	2	TN-CLL, older patients	64	71	ORR	97	19%	not reached (36-month 82%)	NR (36-month 90%)	[52]
Idelalisib + ofatumumab	2	TN-CLL	27	67	ORR	88.9%	1 patient (3.7%)	23 months	NR (36-month 88%)	[53]
Duvelisib	1/IPI 145-02	iNHL, TN CLL, RR CLL T-NHL	210 (75:55 RR; 18 TN)	67	Safety, MTD, PK/PD	56 RR; 83 TN	1 patient (1.8% of RR CLL)	15.7 RR NR TN (12 months—94%)	NR RR (12 months—65.5%) NR TN (12 months—100%)	[54]
Duvelisb	2/DYNAMO	iNHL double refractory to Rtx and to either Cht or RI	129 (28)	65	ORR	67.9	0	ND	ND	[55]
Duvelisib vs. ofatumumab	3/DUO	RR-CLL/SLL	319	69	PFS	74	1 patient (0.6%)	13.3	NR (12 months—86%)	[56]
Umbralisib	1	RR CLL/SLL, RR B-NHL, RR T-NHL, RR HL	90 (24)	65	Safety, MTD, PK	50	0	13.4	ND	[57]
Umbralisib + ublituximab (U2)	1/1b	RR B-NHL, RR CLL/SLL	75 (22)	64	Safety, DLT, MTD	62	ND	27.57	ND	[58]
Umbralisib + ublituximab (U2) vs. O + Chl	3/UNITY-CLL	TN CLL, RR CLL	421 (240 TN; 181 RR)	67	PFS per IRC	83.3	ND	31.9 (38.5 TN; 19.5 RR)	ND	[59]
RETRO-idel	real-world	RR-CLL	83	72	ORR	85.5%	NR	21.7 months	not reached (36-month 55.5%)	[60]
RETRO-idel	real-world	TN-CLL	27	71	ORR	96.3%	NR	18.7 months	not reached (36-month 58.6%)	[60]

B-NHL—B cell non-Hodgkin lymphoma; BR—bendamustine-rituximab; Cht—chemotherapy; CR—complete remission; DTD—dose-limiting toxicity; iNHL—indolent non-Hodgkin lymphoma; IRC—independent review committee; MTD—maximal tolerated dose; ND—no data; NR—not reached; O—ofatumumab; ORR—overall response rate; OS—overall survival; PFS—progression-free survival; R—rituximab; TN-CLL—treatment-naïve chronic lymphocytic leukemia; RR-CLL—relapsed/refractory chronic lymphocytic leukemia; PK pharmacokinetics; PD—pharmacodynamics; Rtx—rituximab; RI—radioimmunotherapy; * data for CLL/SLL cohort; T-NHL—T cell non-Hodgkin lymphoma.

**Table 3 cancers-14-01571-t003:** Summary of non-hematological characteristic PI3Ki-related adverse events across selected studies.

Study	Population	Diarrhea Colitis (All Grade/Grade ≥ 3)	ALT Increase AST Increase (All Grade/Grade ≥ 3)	Infections/Pneumonia (All Grade/Grade ≥ 3)	Pneumonitis (All Grade/Grade ≥ 3)	Cutaneous Adverse Events (All Grade/Grade ≥ 3)	Drug Discontinuation Due to AEs	References
Phase 3 idelalisib + rituximab vs. rituximab (Study 116)	RR-CLL	46.4%/16.4% 10.9%/8.2%	39.1%/9.1% 28.2%/5.5%	NR/32.7% NR/NR	10%/6.4%	10%/3%	5%	[48]
Phase 3 idelalisib + ofatumumab vs. ofatumumab (Study 119)	RR-CLL	53%/19% 6%/6%	53%/11% 37%/9%	78%/NR 19%/14%	6%/5%	18%/2%	39%	[49]
Phase 3 idelalisib + BR vs. placebo + BR	RR-CLL	38%/9.2% 6%/6%	61%/21% 53.5%/15.5%	69%/39% 17%/1%	NR/1.4%	16%/3%	27%	[50]
Phase 2 idelalisib + rituximab in older patients	TN-CLL	64%/42% (diarrhea + colitis)	67%/23% (ALT + AST)	44%/25% 28%/19%	3%/3%	58%/13%	29.7%	[52]
Phase 2 idelalisib + ofatumumab	TN-CLL	46%/17% (diarrhea + colitis)	79%/54% (ALT + AST)	13%/13% NR/NR	13%/8%	NR/R	Study prematurely terminated due to AEs	[53]
Phase 1 Duvelisib (IPI 145 -2 trial)	iNHL, TN CLL, RR CLL, T-NHL	42%/11.4% NR/6%	38.6%/19.3% 37.6%/15.3%	NR/NR 13.3%/9.5%	NR/4%	30.5%/5.7%	NR	[54]
Phase 2, (DYNAMO trial)	iNHL	48.8%/14.7% 7.8%/5.4%	14%/5.4% 10.1%/3.1%	NR/NR	4.7%/NR	NR/NR	31%	[55]
Phase 3, duvelisib vs. ofatumumab (DUO trial)	RR-CLL	51%/15% 13%/12%	NR/3% NR/3%	69%/NR 18/%14%	NR/3%	10%/2%	NR	[56]
Phase 1 umbralisib	RR CLL/SLL, RR B-NHL, RR T-NHL, RR HL	40%/3% 2%/2%	3%/2% 4/%3%	NR/NR 8%/6%	NR/NR	14%/4%	7%	[57]
Phase1/1b umbralisib + ublituximab	RR B-NHL, RR CLL/SLL	64%/0% NR/NR	14%/0% 14%/0%	96%/5% 9%/5%	NR/NR	NR/NR	13%	[58]
Phase 3 umbralisib + ublituximab (UNITY-CLL)	TN CLL, RR CLL	NR/12.1% NR/3.4%	NR/8.3% (ALT + AST)	NR/NR	NR/2.9%	NR/NR	16.5%	[59]

B-NHL—B cell non-Hodgkin lymphoma; BR—bendamustine-rituximab; CR—complete remission; NR—not reported; O—ofatumumab; ORR—overall response rate; OS—overall survival; PFS—progression-free survival; R—rituximab; RR-CLL—relapsed/refractory chronic lymphocytic leukemia; TN-CLL—treatment-naïve chronic lymphocytic leukemia; T-NHL—T cell non-Hodgkin lymphoma; iNHL—indolent non-Hodgkin lymphoma.

**Table 4 cancers-14-01571-t004:** Summary of hematological PI3K related adverse events across selected studies.

Study	Population	Neutropenia * (All Grade/Grade ≥ 3)	Anemia * (All Grade/Grade ≥ 3)	Thrombocytopenia * (All Grade/Grade ≥ 3)	Reference
Phase 3 idelalisib + rituximab vs. rituximab (Study 116)	RR-CLL	30.9%/28.2%	20.9%/9.1%	all grade < 15%	[48]
Phase 3 idelalisib + ofatumumab vs. ofatumumab (Study 119)	RR-CLL	36.0%/35%	23%/14%	14%/11%	[49]
Phase 3 idelalisib + BR vs. placebo + BR	RR-CLL	64%/60%	27%/15%	24%/13%	[50]
Phase 2 idelalisib + rituximab in older patients	TN-CLL	53%/28%	23%/3%	14%/2%	[52]
Phase 2 idelalisib + ofatumumab	TN-CLL	46%/29%	8%/4%	8%/0%	[53]
Phase 1 Duvelisib (IPI 145 -2 trial)	iNHL, TN CLL, RR CLL T-CL	38.6%/20%	24.8%/14.3%	23.3%/14.3%	[54]
Phase 2, (DYNAMO trial)	iNHL	28.7%/24.8%	26.4%/14.7%	18.6%/11.6%	[55]
Phase 3, duvelisib vs. ofatumumab (DUO trial)	RR-CLL	33%/30%	23%/13%	15%/8%	[56]
Phase 1 umbralisib	RR CLL/SLL, RR B-NHL, RR T-NHL, RR HL	14%/13%	15%/9%	10%/6%	[57]
Phase 1/1b umbralisib + ublituximab	RR B-NHL, RR CLL/SLL	55%/50%	9%/5%	9%/0%	[58]
Phase 3 umbralisib + ublituximab (UNITY-CLL)	TN CLL, RR CLL	NR/30.6%	NR	NR	[59]

* data for CLL/SLL cohort; B-NHL—B cell non-Hodgkin lymphoma; BR—bendamustine-rituximab; not reported; iNHL—indolent non-Hodgkin lymphoma; NR—not reported O—ofatumumab; R—rituximab; TN-CLL—treatment-naïve chronic lymphocytic leukemia; RR-CLL—relapsed/refractory chronic lymphocytic leukemia; T-NHL—T cell non-Hodgkin lymphoma.

**Table 5 cancers-14-01571-t005:** Ongoing studies with PI3K inhibitors in monotherapy and novel combinations.

Studied Regimen	Study Phase	Study Population	Study Identifier
Duvelisib in intermittent dosing	2	Relapsed/refractory CLL/SLL	NCT03961672
Duvelisb + venetoclax	1/2	Relapsed/refractory CLL/SLL Richter syndrome	NCT03534323
Umbralisib	2	Treatment-naïve CLL	NCT04163718
Umbralisib + ublituximab added to ibrutinib, acalabrutinib or venetoclax if no MRD is present after at least 6 months treatment	2	Patients with CLL treated with ibrutinib, acalabrutinib or venetoclax	NCT04016805
Umbralisib + ublituximab + venetoclax vs. umbralisib + ublituximab	2/3	Treatment-naïve and previously treated patients with CLL/SLL	NCT03801525
Umbralisib + ublituximab + venetoclax or lenalidomide	1/2	Relapsed refractory CLL/SLL NHL	NCT03379051
Umbralisib + ublituximab	2	Patients with CLL/SLL with progression after novel therapies (BTKi, BCL2i)	NCT04149821
Umbralisib + ublituximab + acalabrutinib	2	Treatment-naïve and previously treated patients with CLL/SLL	NCT04624633
Copanlisib + ibrutinib	2	Relapsed refractory CLL/SLL	NCT04685915
zandelisib (ME-401) ± rituximab ± zanubrutinib	1	Relapsed/Refractory CLL/SLL or B cell NHL	NCT02914938
Parsaclisib + tafasitamab	1/2	Relapsed refractory CLL/SLL Relapsed refractory NHL	NCT04809467
BGB-10188 ± zanubrutinib or ± tislelizumab	1/2	Relapsed refractory CLL/SLL Relapsed refractory NHL Advanced solid tumors	NCT04282018
ACP 319 + acalabrutinib	1	Relapsed refractory CLL	NCT02157324
TQ-B3525	1/2	Relapsed refractory CLL/SLL	NCT04808570

CLL—chronic lymphocytic leukemia; SLL—small lymphocytic lymphoma; BTKi—Bruton’s tyrosine kinase inhibitors; BCL2—B cell lymphoma 2 inhibitor.

## Data Availability

Not applicable.

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
