# Peer review of "PI3K Inhibitors for the Treatment of Chronic Lymphocytic Leukemia: Current Status and Future Perspectives"

_cancers, 2022, doi:10.3390/cancers14061571_

Round 1
Reviewer 1 Report
Overall, this review is well written, and worth publishing in the special issue The present review provides a detailed and broad overview of completed or ongoing clinical trials of the different PI3K inhibitors for CLL therapy. In addition a comparison of the different adverse effects caused by the different inhibitors is reported, further suggesting the potential use of novels agents with a distinct and better safety profile in the next future.
However, several minor errors or typos below described should be corrected before publication and a few references should be added:
Paragraph 3, line 141: please correct the title
Line 146: eliminate 3.1.1
Line 486 : AE @???
Paragraph 3.3: Here it is introduced the novel inhibitor Umbralisib which” has a different chemical structure….”. I suggest to cite the paper from Guarante V, Sporteletti P. Cancer 2021, 13, which show in a figure the 3D structure of Idelalisib, Duvelasib and Umbralisib, or alternatively The Authors may introduce a new figure.
Paragraph 4.1, line 616: Eurpean
Paragraph 4.4, line 658: dmeostrate
Paragraph 4.5.: I suggest to cite specific references describing “the antitumor activity in a T cell leukemia xenograft model…..”
Paragraph 4.7, line 687: patients
Paragraph 4.8: Please introduce reference for “SHC014748M demonstrated promising preclinical antitumor activity in B-cell NHL…..” (probably in Neoplasia 2020)
Paragraph 5, line 743: scessful
Author Response
Reviewer 1.
Comments and Suggestions for Authors
Overall, this review is well written, and worth publishing in the special issue. The present review provides a detailed and broad overview of completed or ongoing clinical trials of the different PI3K inhibitors for CLL therapy. In addition, a comparison of the different adverse effects caused by the different inhibitors is reported, further suggesting the potential use of novels agents with a distinct and better safety profile in the next future. However, several minor errors or typos below described should be corrected before publication and a few references should be added
Response: We thank the Reviewer for positive review of our paper.
Paragraph 3, line 141: please correct the title
Response: Corrected
Line 146: eliminate 3.1.1
Response: Eliminated
Line 486: AE @???
Response: Corrected for “³”
Paragraph 3.3: Here it is introduced the novel inhibitor Umbralisib which” has a different chemical structure….”. I suggest to cite the paper from Guarante V, Sporteletti P. Cancer 2021, 13, which show in a figure the 3D structure of Idelalisib, Duvelasib and Umbralisib, or alternatively The Authors may introduce a new figure.
Response: The paper is cited as requested.
Paragraph 4.1, line 616: Eurpean
Response: Corrected
Paragraph 4.4, line 658: dmeostrate
Response: Corrected
Paragraph 4.5.: I suggest to cite specific references describing “the antitumor activity in a T cell leukemia xenograft model…..”
Response: We cannot identified more specific reference than “Lewis J, Girardi M, Vakkalanka S, et al. RP6530, a Dual PI3Kδ/γ Inhibitor, Attenutates AKT Phosphorylation and Induces Apoptosis In Primary Cutaneous T Cell Lymphoma (CTCL) Cells. Blood. 2013; 122: 4418-4418”
Paragraph 4.7, line 687: patients
Response: Corrected
Paragraph 4.8: Please introduce reference for “SHC014748M demonstrated promising preclinical antitumor activity in B-cell NHL…..” (probably in Neoplasia 2020)
Response: Reference is added
Paragraph 5, line 743: scessful
Response: Corrected
Reviewer 2 Report
The review is providing insight on the mechanism of action and therapeutic role of PI3K inhibitors. Idelalisib, the first-in class PI3Ki that is selective for PI3kdelta isoform showed efficacy in patients with CLL with unfavorable disease features including del(17p)/TP53 mutation, unmutated IGHV and refractory/relapsed disease. In a relevant proportion of cases treatment is complicated by autoimmune-mediated colitis, more rarely pneumonitis, AST/ALT elevation infections and skin changes that limited its widespread use. Duvelisib (a dual PI3Kdelta-gamma inhibitor) is approved in US for use in patients with R/R CLL but showed a similar toxicity profile. Umbralisib, a highly-selective inhibitor of PI3Kdelta and casein kinase-1epsilon (CK1ε), is currently investigated in monotherapy and in combination regimens in phase 3 trials for CLL and iNHL. Novel PI3Kis are in development in early phase clinical trials.
The review is very comprehensive and well-written, I have only few comments and suggestions.
Some minor stylistic comments:
IGVH should be replaced by IGHV without italic
TP53 should be italicized throughout the text
acute caught (page 10 row 332): I'm wondering if "caught" is the right term here, please clarify
relapse/refractory patients: I would suggest to use "relapsed/refractory" instead
"OR was not reached in either cohort" page 11 row 411: I'm not sure I've got the meaning of the abbreviation here, are the authors referring to median OS? Please clarify
"Unmutated MRD" page 15 row 565: it should read undetectable MRD, please double-check
Please check carefully the last sections as there are few typos left
I would consider to replace thorughout the text "CLL patients" with "patients with CLL"
Few comments on the content:
"In the event of progression, patients on rituximab monotherapy could cross over to the R-idelalisib arm within the extension trial" (page 5 row 193): please double-check as the crossover was allowed to idelalisib monotherapy and not idelalisib+rituximab
"Therefore, intermittent dosing of idelalisib, which would allow Treg recovery during the off-drug period, is a plausible option" (page 11 row 388): to the best of my knowledge this has not been tested in clinical trials, I would suggest that the authors provide a reference or soften the sentence as it seems an indication for clinical practice.
Author Response
Reviewer 2
The review is providing insight on the mechanism of action and therapeutic role of PI3K inhibitors. Idelalisib, the first-in class PI3Ki that is selective for PI3kdelta isoform showed efficacy in patients with CLL with unfavorable disease features including del(17p)/TP53 mutation, unmutated IGHV and refractory/relapsed disease. In a relevant proportion of cases treatment is complicated by autoimmune-mediated colitis, more rarely pneumonitis, AST/ALT elevation infections and skin changes that limited its widespread use. Duvelisib (a dual PI3Kdelta-gamma inhibitor) is approved in US for use in patients with R/R CLL but showed a similar toxicity profile. Umbralisib, a highly-selective inhibitor of PI3Kdelta and casein kinase-1epsilon (CK1ε), is currently investigated in monotherapy and in combination regimens in phase 3 trials for CLL and iNHL. Novel PI3Kis are in development in early phase clinical trials.
The review is very comprehensive and well-written, I have only few comments and suggestions.
Some minor stylistic comments:
IGVH should be replaced by IGHV without italic
Response: Repleaced
TP53 should be italicized throughout the text
Response: Italicized as requested
acute caught (page 10 row 332): I'm wondering if "caught" is the right term here, please clarify
Response: The term was corrected. It should be cough.
relapse/refractory patients: I would suggest to use "relapsed/refractory" instead
Response: Repleaced as requested
"OR was not reached in either cohort" page 11 row 411: I'm not sure I've got the meaning of the abbreviation here, are the authors referring to median OS? Please clarify
Response: Yes it should be OS. Corrected.
"Unmutated MRD" page 15 row 565: it should read undetectable MRD, please double-check
Response: Corrected accordingly
Please check carefully the last sections as there are few typos left
Response: Corrected accordingly
I would consider to replace thorughout the text "CLL patients" with "patients with CLL"
Response: Repleaced accordingly
Few comments on the content:
"In the event of progression, patients on rituximab monotherapy could cross over to the R-idelalisib arm within the extension trial" (page 5 row 193): please double-check as the crossover was allowed to idelalisib monotherapy and not idelalisib+rituximab
Response: Reedited as requested
"Therefore, intermittent dosing of idelalisib, which would allow Treg recovery during the off-drug period, is a plausible option" (page 11 row 388): to the best of my knowledge this has not been tested in clinical trials, I would suggest that the authors provide a reference or soften the sentence as it seems an indication for clinical practice.
Response: We softened the tone („Therefore, intermittent dosing of idelalisib, which would allow Treg recovery during the off-drug period, is a plausible option, howver it was not so far tested in a clinical trial setting”)
Reviewer 3 Report
This is a comprehensive review of PI3K inhibitors for the treatment of CLL. The review is well organized and written. It is helpful to a reader trying to get an overview of this field.
I noted a couple of errors in the manuscript that need to be corrected. 1) Line 131: phopsphates is a typo, and it should be phosphatases. 2) The subtitle" 3. PI3K pathway in normal B lymphocytes and CLL cells" is incorrect and needs to be rephrased.
There may be others that can be caught by careful proofreading. But overall, this is a good review article.
Author Response
Reviewer 3
Comments and Suggestions for Authors
This is a comprehensive review of PI3K inhibitors for the treatment of CLL. The review is well organized and written. It is helpful to a reader trying to get an overview of this field.
I noted a couple of errors in the manuscript that need to be corrected.
Response: We thank the Reviewer for positive review of our paper .
We corrected all mistakes indicated by the Reviewer
1) Line 131: phopsphates is a typo, and it should be phosphatases.
Corrected
2) The subtitle" 3. PI3K pathway in normal B lymphocytes and CLL cells" is incorrect and needs to be rephrased.
Rephrased for “Approved PI3K inhibitors for CLL and lymphoid malignancies treatment "
There may be others that can be caught by careful proofreading. But overall, this is a good review article